# Phytochemicals Block Glucose Utilization and Lipid Synthesis to Counteract Metabolic Reprogramming in Cancer Cells

**Qiong Wu, Bo Zhao, Guangchao Sui \*** and **Jinming Shi \***

Key Laboratory of Saline-Alkali Vegetation Ecology Restoration, Ministry of Education, College of Life Science, Northeast Forestry University, Harbin 150040, China; wuqiong@nefu.edu.cn (Q.W.); zhaobo1987@nefu.edu.cn (B.Z.)

\* Correspondence: gcsui@nefu.edu.cn (G.S.); jmshi@nefu.edu.cn (J.S.); Tel.: +86-451-8219-1081 (G.S. & J.S.)

**Abstract:** Aberrant metabolism is one of the hallmarks of cancers. The contributions of dysregulated metabolism to cancer development, such as tumor cell survival, metastasis and drug resistance, have been extensively characterized. "Reprogrammed" metabolic pathways in cancer cells are mainly represented by excessive glucose consumption and hyperactive de novo lipogenesis. Natural compounds with anticancer activities are constantly being demonstrated to target metabolic processes, such as glucose transport, aerobic glycolysis, fatty acid synthesis and desaturation. However, their molecular targets and underlying anticancer mechanisms remain largely unclear or controversial. Mounting evidence indicated that these natural compounds could modulate the expression of key regulatory enzymes in various metabolic pathways at transcriptional and translational levels. Meanwhile, natural compounds could also inhibit the activities of these enzymes by acting as substrate analogs or altering their protein conformations. The actions of natural compounds in the crosstalk between metabolism modulation and cancer cell destiny have become increasingly attractive. In this review, we summarize the activities of natural small molecules in inhibiting key enzymes of metabolic pathways. We illustrate the structural characteristics of these compounds at the molecular level as either inhibitor of various enzymes or regulators of metabolic pathways in cancer cells. Our ultimate goal is to both facilitate the clinical application of natural compounds in cancer therapies and promote the development of novel anticancer therapeutics.

**Keywords:** phytochemical; cancer; metabolic reprogramming; glycolysis; lipogenesis



## 1. Introduction

Cancers are characterized by disordered metabolism. Specific metabolic activities essential to cell transformation or other related biological processes promote tumor growth [1,2]. In order to meet the needs of both energy and biosynthesis, cancer cells can "reprogram" their metabolism systems mainly through the following mechanisms:

(1) Increased uptake and utilization of nutrient substrates, especially glucose. In cancer cells, the glycolytic rate is approximately 30-fold higher than that in normal cells. Thus, the rate-limiting step in glucose metabolism is glucose uptake across the plasma membrane, which is carried out by glucose transporter 1 (GLUT1), a protein with an oncogenic role and frequent overexpression in cancer cells [3,4];

(2) Employing metabolic pathways beneficial to biosynthesis in the catabolism of nutrients. A well-cited example is the "Warburg effect" [5] that reveals the propensity of cancer cells to consume glucose through anaerobic glycolysis even under aerobic conditions. In terms of ATP production, glycolysis is inefficient compared to oxidative phosphorylation. Importantly, converting glucose into lactate instead of utilizing it through mitochondrial metabolism provides more substrates and thus permits the synthesis of various biomolecules, such as fatty acids and amino acids [5]. The pentose phosphate pathway (PPP) is another major pathway of glucose catabolism that produces important resources for biosynthesis, such as ribose 5-phosphate (R5P) and NADPH [6];

(3) Aberrantly activated biosynthesis pathways. To meet the excessive demands of cell membranes and signaling molecules in tumors, a number of pathways related to fatty acid biosynthesis and their desaturation pathways in tumor cells are remarkably active. Fatty acid synthase (FASN), a key enzyme to catalyze the cellular process of fatty acid synthesis, exhibits oncogenic activity and is a bona fide target in cancer therapies [7]. "Reprogrammed" metabolic processes are not only important for cancer cell growth but also increasingly appreciated as a major determinant of cell destiny [8,9]. Rapid proliferation is a key characteristic of cancer cells, which confers them with oncogene addiction and non-oncogene dependence, including metabolic dependence. This allows us to design effective strategies and develop new clinical anticancer agents [10].

Compared with synthetic chemicals, natural compounds have several advantages, such as decent safety, low side effects and multistep targeting; thus, many of them have been used as therapeutic and preventive agents for a number of diseases, including cancers. Various compounds from edible plants and traditional Chinese herbals have been revealed to have suppressive effects on the initiation, development and metastasis of human cancers [11,12]. Some of these compounds with well-characterized anticancer activities, such as paclitaxel, are already in the clinic. Although a number of natural compounds have been used for many years and their anticancer characteristics are well-appreciated, the underlying molecular mechanisms of their tumor-suppressive action still remain elusive.

To date, an increasing number of natural compounds have been demonstrated to block metabolic processes in cancer cells. Many of them can repress tumor growth through directly or indirectly regulating different rate-limiting enzymes, especially those involved in glucose and lipid metabolism [13,14]. For example, GLUT1 is inhibited by multiple components in green tea extracts, and each compound exhibits unique characteristics to exert the activity, dependent on its specific structural feature and GLUT1-binding mode. In addition, as a key enzyme of lipid synthesis, FASN is blocked by many small molecule compounds through their binding to different functional domains.

Comprehensively reviewing and comparing the existing research data can improve our understanding of anticancer mechanisms of natural small molecule compounds. Furthermore, the knowledge of the mechanistic action of natural compounds will provide insights in designing combinatorial medication among these molecules or with other anticancer drugs to improve the effectiveness of cancer therapies.

## 2. Natural Polyphenols Directly Inhibiting Transmembrane Glucose Transport

Compared to normal cells, malignant cells require more glucose as both an energy supply and a resource for biosynthesis. To keep a constant and sufficient supply, glucose transporters, especially GLUT1 (Figure 1), are frequently overexpressed in different types of cancer cells [4,15]. Therefore, targeting glucose transporters represents an effective approach to cancer therapies [16].

### 2.1. Green Tea Extracts with Inhibitory Effects on Glucose Uptake and Output

Catechins, including epicatechin gallate (ECG) and epigallocatechin gallate (EGCG), are major active green tea polyphenols with remarkable inhibitory activity against GLUT1 in cancer cells (Table 1) [17,18].

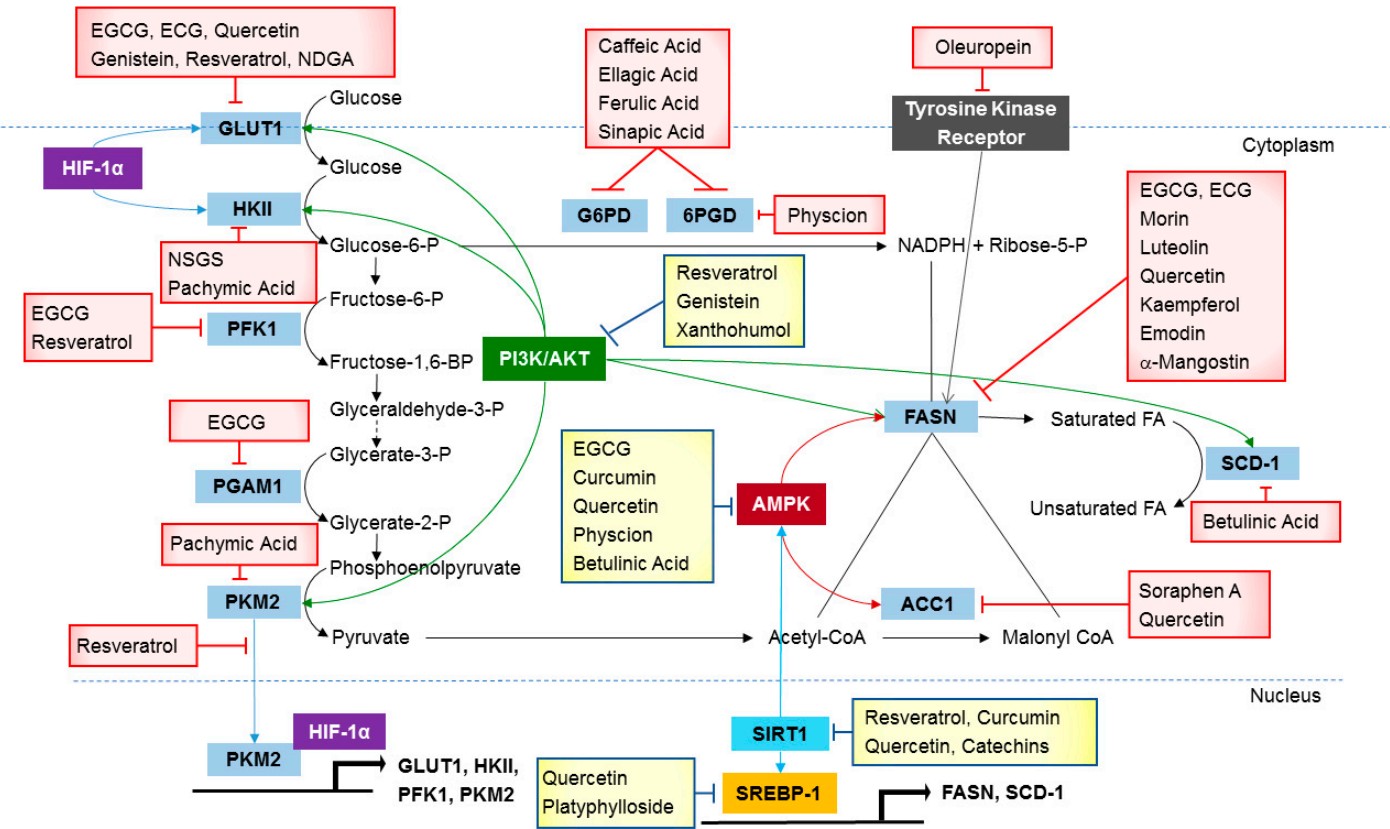

**Figure 1.** Metabolic pathways active in cancer cells are directly inhibited by naturally occurring compounds.

ECG and EGCG inhibit GLUT1 activity through direct binding to the transporter [31,32]. Araujo et al. reported that green tea catechins could significantly reduce the enzymatic efficiency of GLUT1 in human choriocarcinoma BeWo cells [55]. Using three-dimensional topological analysis, Naftalin et al. discovered a putative binding site of the green tea catechins on the intracellular side of the transmembrane protein GLUT1 [31]. This binding site consists of amino acids Arg126-Thr30-Asn288 and Arg126-Thr30-Asn29 that are close to each other in the three-dimensional structure of GLUT1. A more recent study further revealed that the binding region of the three green tea catechins has a hydrophobic cavity consisting of residues Ile164, Val165, Ile168, Phe291, Phe379 and Glu380 [56].

The binding of green tea polyphenols to GLUT1 could alter its substrate recognition in either a competitive or non-competitive manner. For the inward transport, the association of ECG, EGCG and quercetin (Figure 2) with the extracellular side of GLUT1 could competitively block glucose binding to GLUT1; thus, these catechins are competitive inhibitors of glucose uptake [56–58]. In contrast, for the outward transport of glucose, the green tea polyphenols quercetin was demonstrated to exert non-competitive inhibition through attaching to GLUT1, overlapping with the binding cavity of cytochalasin B (CB), another GLUT1 inhibitor [56]. Kinetic analysis also showed that green tea polyphenols decreased both the Km and Vmax values of GLUT1 [55], implicating that its modulation was unlikely through simple competitive or non-competitive inhibition.

**Table 1.** Naturally occurring compounds reported as metabolic antagonists in cancer glucose utilization and lipogenesis. Discussed in this review.

| Compounds | Targets | Biological Models | Solvent | Dosage | References |
|---|---|---|---|---|---|
| Alpha-mangostin | FASN | Breast cancer MCF-7 cells | - | $IC_{50}$:3.57 | [19] |
| | | Breast cancer MDA-MB-231 cells | | $IC_{50}$:3.35 | |
| Betulinic acid | Stearoyl-CoA desaturase 1 | HeLa cells | DMSO | 5–10 μg/mL | [20] |
| | | Colon cancer stem cells | - | - | [21] |
| | AMP-activated kinase pathway | WS-1, A549, MCF-7, H1299, H460 and MDA-MB-231 cells | DMSO | 0–50 μg/mL | [22] |
| Caffeic acid | Glucose-6-phosphate dehydrogenase 6-phosphogluconate dehydrogenase | In vitro | Tris·HCl buffer | $IC_{50}$:0.481, 0.486 mM | [23] |
| | Glucose-6-phosphate dehydrogenase | Cultured rainbow trout gill cells | - | 0–0.1 mM | [24] |
| Cerulenin | FASN (cysteine in β-ketoacyl synthase domain) | In vitro | Potassium phosphate buffer | 0–80 μM | [25] |
| Curcumin | AMP-activated kinase pathway | Ovarian cancer CaOV3 cells | - | 10–50 μM | [26] |
| Ellagic acid | Glucose-6-phosphate dehydrogenase 6-phosphogluconate dehydrogenase | In vitro | Tris/HCl buffer | $IC_{50}$:0.072, 0.188 mM | [23] |
| Emodin | FASN | Colon cancer HCT116 and SW480 cells | - | Emodin (10–50 μM) and/or cerulenin (100 μM) | [27] |
| | Glucose transporter 1 Hexokinase II Phosphofructokinase 1 | Pancreatic cancer MiaPaCa2 cells, Athymic mice carrying pancreatic cancer cells | - | 0–200 μM | [28] |
| Epigallocatechin-3-gallate | FASN | In vitro | - | 0.1–0.35 mM | [29] |
| | | Hepatocellular carcinoma HepG2 and Hep3B cells | DMSO | 0–160 μM | [30] |
| | Acetyl-CoA carboxylase | Hepatocellular carcinoma HepG2 and Hep3B cells | DMSO | 0–160 μM | [30] |
| | Glucose transporter family | Breast cancer MCF-7 and MDA-MB-231 cells | DMSO | 0–100 μM | [17] |
| | | Human intestinal Caco-2/TC7 cells | DMSO | $IC_{50}$:0.091 mg/mL | [18] |
| | | Human erythrocytes | - | $Ki_{EGCG}$: 0.977 μM | [31] |
| | | Choriocarcinoma BeWo cells | - | 0–100 μM | [32] |

**Table 1.** *Cont.*

| Compounds | Targets | Biological Models | Solvent | Dosage | References |
|---|---|---|---|---|---|
| | Phosphofructokinase 1 | HCC-LM3 and HepG2 cells | Phosphate buffer saline | 0–400 μM | [33] |
| | Phosphoglycerate mutase 1 | NCI-H1299 and MDA-MB-231 cells | - | 0–100 μM | [34] |
| | AMP-activated kinase pathway | Hepatocellular carcinoma HepG2 and Hep3B cells | DMSO | 0–160 μM | [30] |
| Genistein | Glucose transporter 1 Hexokinase II | Hepatocellular carcinoma HCC-LM3 and Bel-7402 cells Mouse subcutaneously injected HCC-LM3 cells | DMSO | 0–80 μM | [35] |
| | PI3K/AKT/mTOR signaling pathway | Human intrahepatic CCA HuCCA-1 and RMCCA-1 cells | DMSO | 50–200 μM | [36] |
| | | Human lung adenocarcinoma H460 cells | DMSO | 100 μM | [37] |
| Kaempferol | FASN | In vitro | DMSO | $IC_{50}$:10.38 μM | [38] |
| | | In vitro | DMSO | $IC_{50}$:2.52 μM | [38] |
| Luteolin | FASN | Breast cancer MDA-MB-231 cells and prostate cancer LNCaP cells | DMSO | 0–50 μM | [39] |
| Morin | FASN | In vitro | DMSO | $IC_{50}$:2.33 μM | [38] |
| Oleuropein | Tyrosine kinase signaling pathway | Breast cancer MCF-7 and SKBR3 cells | - | 50 μM | [40] |
| Pachymic acid | Pyruvate kinase M2 Hexokinase II | Breast cancer SKBR-3 cells | DMSO | 0–100 μM | [41] |
| | 6-phosphogluconate dehydrogenase | Lung cancer H1299 and leukemia K562 cells, leukemia cells isolated from PB samples from a representative B-ALL patient. | DMSO | 0–40 μM | [42] |
| Physcion | | Breast cancer MCF-7 and MDA-MB-231 cells | DMSO | 0–40 μM | [43] |
| | AMP-activated kinase pathway | Breast cancer MCF-7 and MDA-MB-231 cells | DMSO | 0–40 μM | [43] |
| Platyphylloside | FASN Stearoyl-CoA desaturase 1 | Mouse 3T3-L1 preadipocytes | DMSO | 0–100 μM | [44] |
| Quercetin | Glucose transporter family | Breast cancer MCF-7 and MDA-MB-231 cells | DMSO | 10–100 μM | [17] |
| | | Choriocarcinoma BeWo cells | - | 0–100 μM | [32] |

<div align="center">**Table 1.** *Cont.*</div>

| Compounds | Targets | Biological Models | Solvent | Dosage | References |
|---|---|---|---|---|---|
| | | In vitro | DMSO | $IC_{50}$:4.29 μM | [38] |
| | FASN | Breast cancer MDA-MB-231 cells and prostate cancer LNCaP cells | DMSO | 0–50 μM | [39] |
| | Acetyl-CoA carboxylase | Rat hepatocytes | DMSO | 0–50 μM | [45] |
| | AMP-activated kinase pathway | Mouse 3T3-L1 preadipocytes | DMSO | 0–100 μM | [46] |
| | FASN | Breast cancer SKBR-3 cells | DMSO | 0–150 μM | [47] |
| | Pyruvate kinase M2 | Cervical cancer HeLa cells, Breast cancer MCF-7 cells, Hepatocellular carcinoma HepG2 cells | DMSO | 50 μM | [48] |
| | Phosphofructokinase 1 | Breast cancer MCF-7 cells | DMSO | 0–100 μM | [49] |
| Resveratrol | Glucose transporter 1 | Ovarian cancer PA-1, OVCAR3, MDAH2774, and SKOV3 cells | DMSO | 50 μM | [50] |
| | | Leukemic *U*-937 and HL-60 cells | DMSO | $IC_{50}$:30 μM | [51] |
| | | Ovarian cancer SKOV3 and CaOV3 cells | DMSO | 0–100 μM | [52] |
| | PI3K/AKT/mTOR signaling pathway | Ovarian cancer PA-1, OVCAR3, MDAH2774, and SKOV3 cells | DMSO | 50 μM | [50] |
| | | Breast cancer SKBR-3 cells | DMSO | 0–150 μM | [47] |
| Rhein | Glucose transporter 1 Hexokinase II Phosphofructokinase 1 | Pancreatic cancer MiaPaCa2 cells, Athymic mice carrying pancreatic cancer cells | - | 0–200 μM | [28] |
| Soraphen A | Acetyl-CoA carboxylase 1 | In vitro | Methanol | 0–54.5 μg/mL | [53] |
| Xanthohumol | PI3K/AKT-GSK3beta-FBW7 signaling pathway | Human glioblastoma U87-MG, T98G and LN229 cells | DMSO | 0–10 μM | [54] |

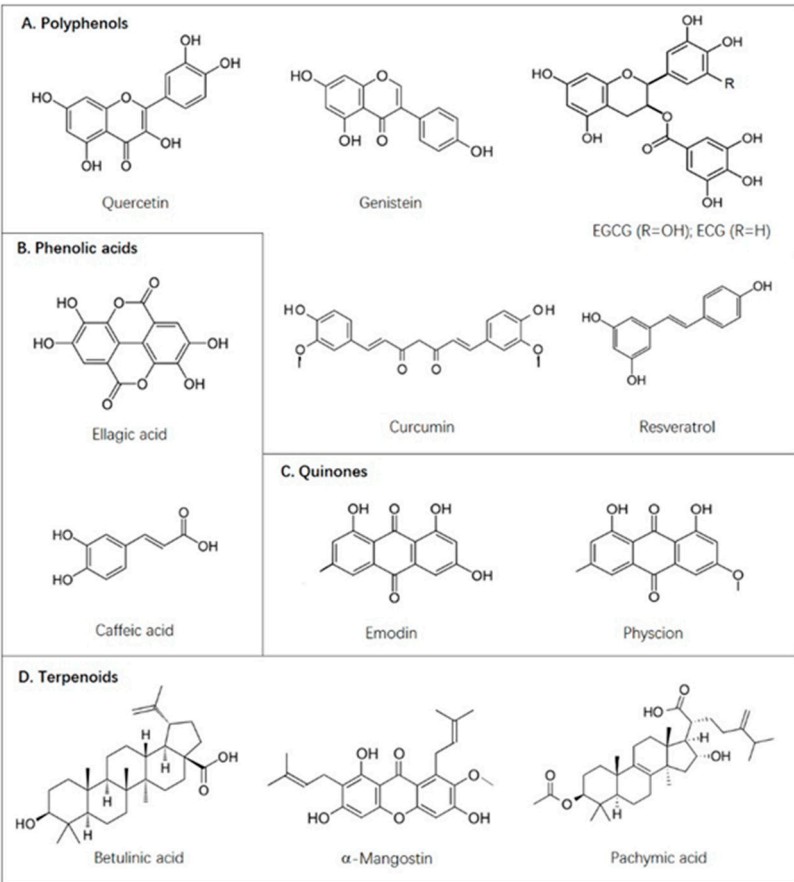

**Figure 2.** Naturally occurring compounds discussed in this paper.

### 2.2. Other Natural Polyphenols with Direct GLUT1-Binding Activities

Besides the well-studied green tea extracts, several other compounds have also been reported to show inhibitory effects on GLUT1. Genistein, a soybean-derived isoflavonoid compound with well-known preventive activity against breast and prostate cancers, can block GLUT1 activity by attaching to its external glucose-binding site [57]. Like green tea polyphenols, genistein behaved as a competitive inhibitor of uptake transport and a non-competitive inhibitor of net sugar output of human red cells, as demonstrated by kinetic analyses [57]. Interestingly, genistein and green tea polyphenols could bind to GLUT1 in different conformations. While genistein attached to GLUT1 with its external surface associating with glucose, green tea polyphenols could bind to this transporter with its cytoplasmic side bound by glucose [57].

Resveratrol is a kind of stilbene extracted from grapes and has attracted strikingly increased attention due to its potent antioxidant, radical scavenging, chemopreventive and anticancer activities [59]. Resveratrol has been considered as a bona fide inhibitor of GLUT1 in multiple studies [60]. Salas et al. reported that resveratrol could block GLUT1-mediated glucose transport through directly binding to one of its internal domains in a non-competitive manner to reduce glucose uptake in human leukemic cells [51]. In human red blood cells, nordihydroguaiaretic acid (NDGA), a natural compound with structural similarity to resveratrol, could also inhibit glucose uptake in a non-competitive manner by attaching to the non-substrate binding site of GLUT1 [13].

### 3. Natural Compounds Inhibiting De Novo Fatty Acid Synthesis and Modification

FASN and acetyl-CoA carboxylase (ACC) are the major enzymes in de novo lipogenesis. Both are overexpressed in multiple types of human cancers and recognized as potential targets in developing novel cancer therapeutic agents [7,61].

### 3.1. Green Tea Extracts with a Galloyl Moiety Inhibit FASN by Competing with NADPH

FASN is a multifunctional enzyme with seven catalytic activities. Among them, the β-ketoacyl reductase (KR) domain and enoyl reductase (ER) domain catalyze NADPH oxidation [62]. EGCG could inhibit the enzymatic activity of FASN in vitro through competing with NADPH to bind the KR domain [29,63]. Based on structural similarity, EGCG likely exerted this function as a competitive inhibitor of NADPH. Importantly, unlike EGCG, other green tea extracts without the galloyl moiety exhibited much-reduced activity in inhibiting FASN, suggesting that this structural feature of EGCG played a critical role in its inhibitory activity [64].

Several well-known FASN inhibitors also can bind to the KR domain of FASN, such as cerulenin and a synthetic inhibitor C75. However, they have been demonstrated to act as non-competitive inhibitors due to their covalent inhibition of the ketoacyl synthase [25]. Thus, the effects of this type of irreversible inhibitor are different from the reversible inhibition of FASN by green tea extracts. Actually, the reversible inhibition of EGCG may be one of the major reasons for its lack of adverse effects when exerting its activities, including modulation of fatty acid synthesis, maintenance of lipid metabolism balance, and prevention of metabolic disorders.

### 3.2. Other Natural Compounds with Inhibitory Activities against FASN

Mechanistic studies of the cancer-preventive effects of flavonoids suggested their activities in blocking fatty acid synthesis [39,65]. An in vitro study using 15 types of flavonoids revealed that 9 compounds, including quercetin and other similar flavonoids such as morin, luteolin and kaempferol, could efficiently inhibit FASN [38]. Another study by Chen et al. also demonstrated that flavonoid extracts from four plants showed significant inhibition of FASN activity when tested in different cancer cell lines [66].

Emodin, a naturally occurring anthraquinone, could inhibit FASN activity [67]. Consistently, this compound also exhibited antiproliferative and proapoptotic activities in cells of different malignancies, including breast cancer, liver cancer, prostate cancer, leukemia and colon cancer [68–72]. A recent study linked the anticancer activity of emodin to its inhibition of FASN [27]. The authors demonstrated that emodin could concurrently suppress FASN activity, downregulate its protein expression, and induce apoptosis of colon cancer cells.

α-mangostin, a natural xanthone extracted from mangosteen pericarp, has a variety of biological functions, including anticancer activity [73,74]. Quan et al. demonstrated that, unlike EGCG, α-mangostin inhibits FASN in a competitive manner regarding acetyl-CoA and a non-competitive manner regarding malonyl-CoA [75]. An intracellular study also demonstrated that α-mangostin could act as a potent inhibitor to downregulate FASN expression and block its activity, leading to reduced levels of intracellular fatty acid and eventually cancer cell apoptosis [19].

### 3.3. Natural Compounds as Acetyl-CoA Carboxylase (ACC) Inhibitors

ACC catalyzes the production of malonyl-CoA, an essential substrate in fatty acid synthesis. Two ACC isoforms have been reported in mammals, i.e., ACC1 and ACC2 (also known as ACCα and ACCβ). In the past decades, ACC inhibitors have been used in various clinical treatments of human diseases, including microbial infections, metabolic syndrome, diabetes and cancers [76,77].

Soraphen A, a polyketide isolated from the myxobacterium *Sorangium cellulosum*, was first identified as a natural molecule with inhibitory activity against ACC1, which is overexpressed in human cancer cells [53,78]. Further studies revealed that Soraphen A could allosterically promote the catalytically inactive conformation of ACC1 through binding to its biotin carboxylase domain with high affinity [79,80]. Quercetin also showed selective inhibition to ACC without any detectable effect on FASN in rat hepatocytes and could reduce the synthesis of both fatty acid and triacylglycerol [45]; however, the underlying mechanism remained unclear.

### 3.4. Betulinic Acid-Mediated Inhibition of Fatty Acid Desaturation

As a lupane-type triterpenoid produced in birch plants, betulinic acid and its derivatives have been demonstrated to have potent anticancer and anti-HIV activities in numerous studies. To date, the biosynthetic pathway of betulinic acid has been completely mapped, and its commercial production is through phytochemical extraction and semi-synthesis from its precursor, betulin [81]. Natural plant-derived betulinic acid exhibited tumor-selective inhibitory activity and could induce apoptosis of a wide variety of cancer cells, mostly through a mitochondrion-dependent mechanism [82–85].

In the fatty acid synthesis network, stearoyl-CoA desaturase (SCD) is an enzyme catalyzing the rate-limiting step in the production of monounsaturated fatty acids, and its overexpression is reportedly associated with poor clinical outcomes of cancer patients [86,87]. Remarkably, inhibition of SCD could more efficiently block cancer cell proliferation than targeting other enzymes in the de novo lipogenesis pathway, such as ACC and FASN [88]. Potze et al. reported that betulinic acid could inhibit SCD activity and increase the saturation level of mitochondrial lipid cardiolipin. The change of cardiolipin spatial structure could further enhance mitochondrial permeability and consequently led to cytochrome c release and mitochondrion-dependent cell death [20]. Later on, these authors also reported that betulinic acid could induce rapid death of colon cancer stem cells through inhibiting SCD1, an isoform highly expressed in multiple malignancies [21].

## 4. Phenolic Acids and Physcion as Inhibitors of the Pentose Phosphate Pathway (PPP)

The PPP is an important pathway of glucose utilization in addition to glycolysis. This pathway is especially critical to cancer cells because it not only generates R5P to supply high rates of de novo nucleotide synthesis but also provides NADPH required for both fatty acid synthesis and cell survival [89]. Consistently, aberrant activation of PPP was frequently observed in various types of cancer cells [90–92].

Phenolic acids are organic chemicals containing phenolic rings extracted from plants. Multiple phenolic acid compounds, including caffeic acid, ellagic acid, ferulic acid and sinapic acid, have been reported to inhibit glucose-6-phosphate dehydrogenase (G6PD) and 6-phosphogluconate dehydrogenase (6PGD), the latter of which is a rate-limiting enzyme of the PPP [23]. Caffeic acid could cause accumulation of the G6PD mRNA in cultured rainbow trout gill cells [24], although the molecular mechanism underlying this inhibition has not been resolved.

As a key enzyme of the PPP, 6PGD plays a key role in the oncogenic process and has been recognized as an effective target in cancer therapies [42]. Physcion was identified as a 6PGD inhibitor from a library of 2000 Food and Drug Administration (FDA)-approved small molecule compounds [42,93]. A recent study by Yang et al. showed that physcion could inhibit 6PGD, but not G6PD, leading to reduced proliferation of human lung cancer and breast cancer cells [43]. Based on the crystal structure of 6PGD, physcion could fit in a pocket near the binding site of glucose-6-phosphate (G6P) and dampen lipogenesis, leading to reduced growth in xenograft tumors of nude mice without causing any detectable side effect [42]. Multiple recent studies verified physcion as a bona fide inhibitor of 6PGD [94–96].

## 5. Natural Compounds Modulating Key Enzymes in Aerobic Glycolysis

Glucose transported to the cytoplasm can be phosphorylated by hexokinases (HK) with ATP as a phospho donor. Among the four isoforms of HK, HKII is frequently highly expressed in malignant cells and plays an important role in tumor initiation and progression [97]. Bao et al. [98] extracted a natural steroid from *Ganoderma sinense* (NSGS) and identified it as the first natural inhibitor of HKII. In their study, a natural product, (22E,24R)-6β-methoxyergosta-7,9(11),22-triene-3β,5α-diol, exhibited high binding affinity to HKII in vitro. Consistently, this compound showed clear inhibitory effects on human pancreatic cancer cells with 4-fold selectivity versus normal cells, suggesting its potential as a candidate drug in the therapies of pancreatic cancer [98].

Pyruvate kinase (PK) catalyzes the final step of glycolysis by converting phospho-enolpyruvate and ADP to pyruvate and ATP [99]. The pyruvate kinase M2 (PKM2) is the major isoform of PK and overexpressed in many types of cancer cells [100]. Pachymic acid, a lanostane-type triterpenoid from Poria cocos, could bind PKM2 in the pocket of its natural activator, fructose-1,6-bisphosphate and block the glycolysis of breast cancer cells. Additionally, pachymic acid is also an inhibitor of HKII [41].

The 6-phosphofructo-1-kinase (PFK) is a rate-limiting enzyme of glycolysis, and its inhibition can lead to breast cancer cell death [101]. Thus, PFK is a potential target in developing novel anticancer therapeutics. Gómez et al. [49] first demonstrated that resveratrol could directly inhibit PFK activity in both breast cancer cells and an in vitro assay by promoting its dissociation from fully active tetramers to low active dimers. Similarly, a study by Li et al. demonstrated that EGCG could also attenuate PFK activity through modulating its oligomeric structure [33].

Phosphoglycerate mutase 1 (PGAM1) is a mutase catalyzing the reversible conversion of 3-phosphoglycerate to 2-phosphoglycerate in the glycolytic pathway and thus coordinates glycolysis and biosynthesis to support tumor growth [102]. EGCG was identified as a PGAM1 inhibitor with remarkably stronger potency than previously reported inhibitors, such as PGMI-004A. A mechanistic study showed that the inhibition of PGAM1 by EGCG was caused by conformational change upon their binding rather than competitive inhibition of the substrate. Through inhibiting PGAM1, EGCG decreased 2-phosphoglycerate production and further inhibited glycolysis and PPP, leading to reduced cancer cell proliferation [34].

## 6. Natural Compounds Inhibiting Protein Expression of Metabolic Enzymes

### 6.1. Downregulation of De Novo Lipogenesis by Activating AMPK

The AMP-activated kinase (AMPK) pathway is one of the most important signaling pathways regulating biological energy metabolism. Activated AMPK represses its downstream target ACC through promoting its phosphorylation and downregulates the genes of multiple lipogenic enzymes [103,104]. Curcumin was reported as an activator of AMPK in a p38-dependent manner and thus could increase the phosphorylation of both AMPK and ACC in cancer cells [26]. EGCG, quercetin and physcion could also promote AMPK-induced ACC phosphorylation and FASN downregulation, leading to markedly reduced endogenous lipogenesis in cancer cells [30,42,43,46]. Additionally, betulinic acid could also activate AMPK and subsequently reduce ACC activity with concomitant inhibition of glucose-mediated lipogenesis [22,105].

### 6.2. Inhibition of Glucose Utilization and Lipogenesis through the PI3K/AKT/mTOR Signaling Pathway

The AKT (also known as protein kinase B, PKB) pathway is an important signaling pathway regulating glucose utilization and lipogenesis. Phosphatidylinositol 3-kinase (PI3K) catalyzes the formation of the second messenger phosphatidyl-inositol-1,4,5-trisphosphate (PIP3) on the plasma membrane, while PIP3 binds to AKT and phosphoroside dependent kinase-1 (PDK1) to promote AKT activation through its phosphorylation at T308. Activated AKT regulates cellular function by phosphorylating downstream effectors, including various enzymes, kinases and transcription factors. Therefore, AKT has been considered as a bona fide target in cancer therapies [106]. As a mammalian target of rapamycin, mTOR plays a key role in metabolic regulation through responding to upstream signals from AKT, sensing cellular material and energy reserves, and mediating the balance of sugar and lipid metabolism. Over the past decade, several studies demonstrated that resveratrol exerted its anticancer activity through attenuating AKT and mTOR regulation [52,107,108]. AKT and mTOR phosphorylation are important signals for increased glucose uptake and glycolysis. Kueck et al. reported that resveratrol could block the phosphorylation of the two proteins, leading to reduced glucose uptake and lactate production, and eventually, the autophagocytosis of ovarian cancer cells [52]. Interestingly, another study indicated that resveratrol could interrupt intracellular GLUT1 trafficking to the plasma membrane,

reduce glucose uptake and eventually induce apoptosis of ovarian cancer cells without altering GLUT1 expression at the mRNA or protein levels [50]. Additionally, resveratrol could also downregulate FASN and HER2 genes to synergistically induce apoptosis of breast cancer cells [47]. The anticancer activity of resveratrol could also take place through PI3K/AKT/mTOR signaling by upregulating PTEN and attenuating AKT activation [47]. Similarly, resveratrol could also inhibit PKM2 expression through dampening mTOR activity in various cancer cells [48]. As a phytochemical compound, genistein has been reported to exhibit a series of anticancer activities, including antagonizing estrogen receptor, blocking epidermal growth factor receptor, and inhibiting AKT [36]. Consistent with its negative regulation in AKT activation, genistein was shown to repress the activities of several lipogenesis-related enzymes activated by AKT, including FASN and SCD-1 [37,109].

In a recent study, xanthohumol, a natural product extracted from hop plant *Humulus lupulus* L., was demonstrated to inhibit HK2 expression by antagonizing the PI3K/AKT-GSK3β-FBW7 signaling axis [54]. Xanthohumol could also inhibit glucose uptake of cells, and its inhibitory activity was remarkably higher than that of resveratrol and green tea polyphenols [110]. These studies implicated great prospects of xanthohumol in both basic research and therapeutic application.

### 6.3. Downregulation of FASN through Activating Tyrosine Kinase Receptor

Oleuropein is present in both the pulp and leaf extracts of olive and is a major phenolic component in the Mediterranean diet. A recent study demonstrated that oleuropein could significantly reduce the viability of breast cancer cells, suggesting its potential as a promising herbal medication to treat cancers [111]. Another report revealed the activity of oleuropein in reducing the expression of GLUT1 and PKM2 in the glycolysis pathway, although the mechanism underlying this regulation was not elucidated [112]. Menendez et al. reported that polyphenols, flavonoids and secoiridoids extracted from olive oil could significantly suppress FASN protein levels in HER2-overexpressing breast cancer cells, including HER2 gene-amplified SKBR3 cells and engineered HER2-overexpressing MCF-7 cells. Their research revealed a novel mechanism that phenolics in olive oil regulated FASN expression through modulating a tyrosine kinase receptor network [40].

### 6.4. Inhibition of FASN through Suppressing SREBP1

As upstream regulators in lipid metabolism signaling pathways, sterol regulatory element-binding proteins (SREBPs) play a key role in FASN gene expression [113]. Quercetin is a natural molecule produced in apples, onions, teas and berries, and possesses antihistamine and anti-inflammatory activities. Seo et al. reported that quercetin could remarkably reduce SREBP1 and FASN expression at both mRNA and protein levels in mouse stromal cells [114]. Similarly, platyphylloside, a diarylheptanoid isolated from *Betula platyphylla*, also inhibited the expression of SREBP1, and its downstream targets FASN and SCD-1, leading to attenuated adipocyte differentiation in mouse embryonic fibroblasts (preadipocytes) [44]. Furthermore, the lipid portion extracted from a special blue-green alga could also decrease FASN and SCD-1 expression through targeting SREBP1 [115].

### 6.5. Inhibition of Glycolysis through Downregulating HIF-1α

As an important anticancer compound, genistein could both sensitize liver cancer cells to apoptosis through downregulating hypoxia-inducible factor-1α (HIF-1α) and suppress aerobic glycolysis through inactivating critical regulators of glucose uptake and activation, including GLUT1 and HKII [35]. Two additional natural compounds, emodin and rhein extracted from *Rheum palmatum* could also downregulate HIF-1α expression to inhibit glycolysis in the studies using both human pancreatic cancer cells and animal models [28].

### 6.6. Inhibition of Histone Deacetylases

For decades, targeting or inhibiting histone deacetylases has been extensively used as an effective strategy in cancer therapies [116]. On the other hand, multiple compounds

could also dampen cancer cell metabolic activities through activating Sirtuin 1 (SIRT1) [117], a histone deacetylase that deacetylates both histones and non-histone proteins, including a number of transcription factors, and thereby regulate a variety of physiological processes, including glucose metabolism and adipogenesis [118]. Polyphenols, including resveratrol, curcumin, quercetin and catechins, have been shown to activate SIRT1 directly or indirectly in different model systems [119]. In hepatocellular carcinoma cells, resveratrol attenuated fat deposition through inhibiting SREBP1 expression via the SIRT1–FOXO1 pathway. Resveratrol could also promote glucose transportation in insulin-resistant adipocytes through the SIRT1–AMPK pathway [120]. A clinical study of diabetic patients revealed that resveratrol treatment induced upregulation of glucose transporters through the SIRT1–AMPK pathway in skeletal muscle [121].

## 7. Resveratrol-Mediated PKM2 Nuclear Translocation

As a key enzyme regulating the final step of glycolysis, PKM2 can stay in two different oligomer statuses in the cytoplasm. When forming dimers with relatively low activity, PKM2 can promote aerobic glycolysis towards anabolism; however, in a tetramer form with high activity, PKM2 facilitates oxidative phosphorylation for ATP production [100]. Additionally, dimeric PKM2 in the nucleus could also mediate the activities of transcription regulators, such as STAT3 and HIF-1$\alpha$. Under hypoxic conditions, PKM2 could directly interact with HIF-1$\alpha$ to promote its binding and p300 recruitment to its binding elements and transactivate the expression of its target genes, including GLUT1, LDHA and PDK1 [122]. Wu et al. demonstrated that resveratrol could block PKM2 nuclear translocation in human endothelial cells and thus inhibit the expression of GLUT1, HKII and PFK, leading to reduced aerobic glycolysis [123].

## 8. Combinations of Compounds in Cancer Treatments

Different compounds may exert anticancer effects through distant mechanisms. Thus, their combinatorial uses represent promising therapeutic strategies. Numerous studies evaluated the combinations among the compounds in Table 1 or their combinations with other molecules in cancer treatments. For example, the cotreatment of curcumin and emodin could cause synergistic effects in inhibiting the proliferation and invasion of breast cancer cells [124]. In addition, emodin also synergistically enhanced the antitumor effects of paclitaxel against lung cancer both in vitro and in vivo [125]. However, compounds targeting the same or tightly related pathways are generally unable to cause synergistic effects when used together. For example, the combination of emodin and cerulenin, both reported as inhibitors of FASN could only generate additive effects on FASN inhibition in colon cancer cells [27]. Therefore, whether the combination of two or more compounds can achieve highly increased anticancer effects depends on the targeted genes or pathways of individual molecules and requires experimental validation.

## 9. Conclusions

Studies on cell-autonomous reprogramming of cancer metabolism uncovered new principles in metabolic regulation and crosstalk among different cell signaling pathways and the metabolic network [126]. Highly proliferative cells require both energy and biosynthesis to replicate the entire cellular contents. Therefore, metabolic activity is increasingly appreciated as a major determinant of cell proliferation [8,9]. Aberrant metabolism of neoplastic cells provides opportunities in developing cancer therapeutics; thus, increasing efforts have been devoted to investigating how to effectively target cancer metabolism [127]. In addition to cancers, several other diseases, such as diabetes, are also associated with metabolic disorders. Altered metabolism can be exploited to identify new targets to improve patient treatments. As we summarized above, many natural compounds can correct or adjust metabolic abnormalities of human diseases. Therefore, it is of great significance to explore the remedying effects of natural compounds to relieve symptoms and provide insightful supports in cancer therapies.

**Funding:** This research was funded by the Fundamental Research Funds for the Central Universities (2572020DY13) to G.S., the National Natural Science Foundation of Heilongjiang, China (LH2020H002) to J.S., and the National Natural Science Foundation of China (81672795 and 81872293) to G.S.

**Institutional Review Board Statement:** Not applicable.

**Informed Consent Statement:** Not applicable.

**Data Availability Statement:** Not applicable.

**Acknowledgments:** The authors would like to thank Daniel B. Stovall (College of Arts and Sciences, Winthrop University, Rock Hill, SC, USA) for critically reading the manuscript.

**Conflicts of Interest:** The authors declare no conflict of interest.

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
