# Peer review of "Phytochemicals Block Glucose Utilization and Lipid Synthesis to Counteract Metabolic Reprogramming in Cancer Cells"

_applsci, doi:10.3390/app11031259_

Round 1

Reviewer 1 Report

In this review, authors summarize many natural compounds that exert anti-metabolic activities in cancer cells. The paper is well synthesized and includes major phytochemicals. But I feel it is missing some key points. Currently there are many reviews available summarizing the potential and utilization of phytochemicals in cancer therapy. Authors should make the abstract and introduction more targeted to further highlight the uniqueness of this review. 

 Additionally, in place of figure 2, titled "Naturally occurring compounds discussed in this paper", it will be helpful to have a summary table listing all the compounds discussed in this paper, their mode of action, current research stage, and references.

Author Response

Comment 1: I am especially interested in asking why you have focused on these natural products. Why haven't you also included other daily products due to their wide (and well-known) distribution in the Mediterranean diet?

Reply: Natural products are the major source of traditional medicines and various generic therapeutics are derived from natural compounds. Many natural compounds, especially these extracted from medicinal plants, possess both cancer preventive and anticancer activities. In a number of studies, natural products exhibited selective inhibition of cancer cells. In this review, we only focused on the natural products antagonizing glucose utilization and lipid synthesis in cancer cells. We discussed the compounds that could act as substrate analogues or non-competitive inhibitors of enzymes in metabolic pathways, or indirectly regulate the expression and modification of the enzymes through different mechanisms. However, restricted by the length of this review, it is impossible to include all natural compounds.

Nevertheless, we went through several articles related to oleuropein and xanthohumol, and felt that the activity of these compounds suited the theme of this review. Therefore, we added discussion of oleuropein and xanthohumol in the revised manuscript.

Comment 2: Do you have evidence of increasing anticancer efficacy by combining the compounds you mention? For each natural compound, I miss the most appropriate doses of each compound alone (and combined if any) and its dissolution medium for administration.

Reply: We thanks the reviewer for raising this interesting and important point of combinatorial uses of the compounds. Actually, there are numerous papers examining the combinations of different natural compounds, or their combinations with other therapeutics, in the treatments of cancers, and it is impossible to discuss this topic too extensively in the current review article. Nevertheless, we added a new section of “Combinations of compounds in cancer treatments” (as section 8) in the revised manuscript to discuss this topic using several examples.

As suggested by this reviewer, we have added the dosages of the compounds (as either IC50 values or administrated concentrations) and their dissolution reagents in Table 1.

Reviewer 2 Report

The authors of this review emphasize the anticancer activities of natural compounds that target aberrant metabolic processes typical of any type of cancer, such as glucose transport, aerobic glycolysis, fatty acid synthesis, and desaturation.

The authors do a remarkable job to outline the binding sites of each of these natural compounds to the catalytic centers (or structures from in silico assays) of the enzymes involved in each metabolic pathway studied.

1) Even so, I am especially interested in asking why you have focused on these natural products. Why haven't you also included other daily products due to their wide (and well-known) distribution in the Mediterranean diet?

I would suggest to the authors to mention other low-cost and non-toxic natural compounds, such as oleuropein and xanthohumol.

Oleuropein is known to be a major phenolic component in the Mediterranean diet, found both in the pulp of olives and in extracts of olive leaves. In addition, its effectiveness against cancer is recognized by inhibiting glycolysis, by decreasing the expression of glucose transporter-1, protein kinase isoform M2, reducing FASN, etc (I-III).

Xanthohumol (XN) is another antioxidant compound in hops, even 200 fold more powerful than resveratrol, which has profound antitumor activity by inhibiting aerobic glycolysis (IV).

Do you have evidence of combining the compounds you mention with oleuropein?

References:

  1. Asgharzade S, Sheikhshabani SH, Ghasempour E, Heidari R, Rahmati S, Mohammadi M, Jazaeri A, Amini-Farsani Z. The effect of oleuropein on apoptotic pathway regulators in breast cancer cells. Eur J Pharmacol. 2020 Nov 5;886:173509. doi: 10.1016/j.ejphar.2020.173509.
  2. Ruzzolini J, Peppicelli S, Bianchini F, Andreucci E, Urciuoli S, Romani A, Tortora K, Caderni G, Nediani C, Calorini L. Cancer Glycolytic Dependence as a New Target of Olive Leaf Extract. Cancers. 2020; 12(2):317. https://doi.org/10.3390/cancers12020317
  3. Menendez JA, Vazquez-Martin A, Oliveras-Ferraros C, Garcia-Villalba R, Carrasco-Pancorbo A, Fernandez-Gutierrez A, Segura-Carretero A. Analyzing effects of extra-virgin olive oil polyphenols on breast cancer-associated fatty acid synthase protein expression using reverse-phase protein microarrays. Int J Mol Med. 2008 Oct;22(4):433-9. PMID: 18813848.
  4. Yuan, J., Peng, G., Xiao, G., Yang, Z., Huang, J., Liu, Q., Yang, Z., & Liu, D. (2020). Xanthohumol suppresses glioblastoma via modulation of Hexokinase 2 -mediated glycolysis. Journal of Cancer, 11(14), 4047–4058. https://doi.org/10.7150/jca.33045

2) Do you have evidence of increasing anticancer efficacy by combining the compounds you mention? For each natural compound, I miss the most appropriate doses of each compound alone (and combined if any) and its dissolution medium for administration.

Author Response

Comment 1: Authors should make the abstract and introduction more targeted to further highlight the uniqueness of this review. 

Reply: As suggested by the reviewer, we revised the Abstract and Introduction to highlight the major goal of this review article, which is to reveal the inhibition or regulation of metabolic pathways in cancers by natural products from a molecular point of view.

Comment 2: It will be helpful to have a summary table listing all the compounds discussed in this paper, their mode of action, current research stage, and references.

Reply: According to the reviewer’s suggestion, we added a table (Table 1 in the revised manuscript) to list all compounds discussed in this manuscript.